# Synthesis and Characterization of Radiogallium-Labeled Cationic Amphiphilic Peptides as Tumor Imaging Agents

**DOI:** 10.3390/cancers13102388

**Published:** 2021-05-14

**Authors:** Takeshi Fuchigami, Takeshi Chiga, Sakura Yoshida, Makoto Oba, Yu Fukushima, Hiromi Inoue, Akari Matsuura, Akira Toriba, Morio Nakayama

**Affiliations:** 1Department of Hygienic Chemistry, Graduate School of Biomedical Sciences, Nagasaki University, 1–14 Bunkyo-machi, Nagasaki 852-8521, Japan; t.baseball.748@gmail.com (T.C.); yoshida-s@nagasaki-u.ac.jp (S.Y.); yuyou1013@yahoo.co.jp (Y.F.); hello.tangled@gmail.com (H.I.); bb30217033@ms.nagasaki-u.ac.jp (A.M.); toriba@nagasaki-u.ac.jp (A.T.); 2Graduate School of Medical Science, Kyoto Prefectural University of Medicine, 1–5 Shimogamohangi-cho, Sakyo-ku, Kyoto 606-0823, Japan; moba@koto.kpu-m.ac.jp

**Keywords:** cationic amphiphilic peptide, cancer, radiogallium, molecular imaging

## Abstract

**Simple Summary:**

Cationic amphiphilic peptides (CAPs), such as SVS-1, exhibit preferential cytotoxicity towards cancer cells. SVS-1 can also be used as a diagnostic tool for the detection of cancer. Here, we report the development of three ^67^Ga-labeled SVS-1 derivatives as potential cancer diagnostic agents. All of these radiotracers showed high accumulation in cancerous KB cells. Notably, the uptake of ^67^Ga-NOTA-KV6 and ^67^Ga-NOTA-HV6 into 3T3-L1 fibroblasts was significantly lower than in KB cells. In vivo biodistribution studies showed that ^67^Ga-NOTA-KV6, ^67^Ga-NOTA-RV6, and ^67^Ga-NOTA-HV6 exhibit high tumor accumulation, tumor/blood ratios (3.8–8.0), and tumor/muscle ratios (3.3–5.0) 120 min after administration. These radiolabeled SVS-1 analogues target the surface membranes of cancer cells, and are prospective scaffolds for in vivo imaging agents.

**Abstract:**

SVS-1 is a cationic amphiphilic peptide (CAP) that exhibits a preferential cytotoxicity towards cancer cells over normal cells. In this study, we developed radiogallium-labeled SVS-1 (^67^Ga-NOTA-KV6), as well as two SVS-1 derivatives, with the repeating KV residues replaced by RV or HV (^67^Ga-NOTA-RV6 and ^67^Ga-NOTA-HV6). All three peptides showed high accumulation in epidermoid carcinoma KB cells (53–143% uptake/mg protein). Though ^67^Ga-NOTA-RV6 showed the highest uptake among the three CAPs, its uptake in 3T3-L1 fibroblasts was just as high, indicating a low selectivity. In contrast, the uptake of ^67^Ga-NOTA-KV6 and ^67^Ga-NOTA-HV6 into 3T3-L1 cells was significantly lower than that in KB cells. An endocytosis inhibition study suggested that the three ^67^Ga-NOTA-CAPs follow distinct pathways for internalization. In the biodistribution study, the tumor uptakes were found to be 4.46%, 4.76%, and 3.18% injected dose/g of tissue (% ID/g) for ^67^Ga-NOTA-KV6, ^67^Ga-NOTA-RV6, and ^67^Ga-NOTA-HV6, respectively, 30 min after administration. Though the radioactivity of these peptides in tumor tissue decreased gradually, ^67^Ga-NOTA-KV6, ^67^Ga-NOTA-RV6, and ^67^Ga-NOTA-HV6 reached high tumor/blood ratios (7.7, 8.0, and 3.8, respectively) and tumor/muscle ratios (5.0, 3.3, and 4.0, respectively) 120 min after administration. ^67^Ga-NOTA-HV6 showed a lower tumor uptake than the two other tracers, but it exhibited very low levels of uptake into peripheral organs. Overall, the replacement of lysine in SVS-1 with other basic amino acids significantly influenced its binding and internalization into cancer cells, as well as its in vivo pharmacokinetic profile. The high accessibility of these peptides to tumors and their ability to target the surface membranes of cancer cells make radiolabeled CAPs excellent candidates for use in tumor theranostics.

## 1. Introduction

Contrary to healthy cells, the surface membrane of cancer cells tends to be negatively charged due to the exposure of phosphatidylserine (PS) on the outer surface of the lipid bilayer [1,2,3], the changes in the composition of glycoprotein sugar chains [4,5], and the increase in sialic acid content [6,7]. Several hydrophobic cationic amphiphilic peptides (CAPs) have been reported to exhibit antitumor activity [8]. Their anticancer effects are thought to be driven by electrostatic interactions between the CAPs and the negatively charged surface membranes of cancer cells. These peptides are less toxic to normal cells, since the cell surface membrane of healthy cells has an overall neutral charge [9]. Therefore, these peptides show great promise as a new class of anticancer agents. CAPs may be used as tumor imaging agents for the non-invasive diagnosis of cancers. Nuclear medicine imaging techniques such as positron emission tomography (PET) and single photon emission computed tomography (SPECT) allow for the ultrasensitive, high-quantity, non-invasive, deep-tissue visualization of tumors [10,11]. However, effective PET and/or SPECT imaging agents derived from CAPs have yet to be explored.

SVS-1 (KVKVKVKV^D^PPTKVKVKVK-NH_2_; ^D^P is d-proline) is an 18-residue cationic amphipathic peptide composed of repeating lysine-valine (KV) residues linked by a β-turn tetrapeptide (−V^D^PPT−). SVS-1 only folds at the negatively charged surface membrane of cancer cells, acquiring a β-hairpin structure that results in the disruption of the cancer cell membrane while leaving normal cells unaffected [12]. It has been reported that SVS-1 may also be used as a carrier for the targeted intracellular delivery of anticancer agents [13] or nanoparticles [14,15] to cancer cells. The selective anticancer and membrane-interactive properties of SVS-1 make it a good lead peptide for the development of nuclear medicine imaging probes for cancer. Since the KV repeat sequence is directly involved in cancer cell invasion, its length is likely to significantly affect its binding affinity and selectivity for cancer cells [12,16,17]. The linking of two KV residue segments via the β-turn tetrapeptide (−VDPPT−) is important for the formation of the folded β-hairpin on the surface of cancer cell membranes [12]. Therefore, the first part of our study involved exploring the effects of varying the length of the KV repeat segments. Since SVS-1 has six KV sequences, it is referred to as KV6. We designed new SVS-1 derivatives with shorter and longer sequences of KV repeats (KV4 and KV8, respectively). Medina et al. reported the synthesis of CLIP6 (KVRVRVRV^D^PPTRVRERVK-NH_2_), a KV6 derivative with the six lysine residues replaced by arginine, and one valine residue replaced by a glutamic acid [18]. Like KV6, CLIP6 exhibits high membrane permeability and selective toxicity towards cancer cells [18,19]. This suggests that replacing the lysine residues in KV6 with other amino acids may lead to the development of highly selective derivatives with diverse physiological properties. The histamine-containing peptide ^68^Ga-IMP-288 has been used for the pre-targeted PET imaging of carcinoembryonic antigen, which exhibited a quite low uptake in normal tissues [20,21]. In addition, several histidine-rich peptides have also been shown to exhibit high cell permeability and tumor accumulation properties [22,23]. Therefore, we also designed novel peptides in which the lysine residues of KV6 were replaced by arginine or histidine (RV6 and HV6, respectively), as shown in Figure 1.

Radiogallium is widely used in nuclear medicine imaging (^68^Ga for PET and ^67^Ga for SPECT). ^68^Ga (half-life 68 min) is a versatile PET nuclide that can be manufactured using a ^68^Ge-^68^Ga generator without a cyclotron [24,25,26]. In this study, we used ^67^Ga (half-life of 78 h) for radiolabeling and primary biological evaluation because it is easier to handle. In this study, we synthesized and evaluated several fluorescein isothiocyanate isomer I (FITC)- and ^67^Ga-labeled KV6 derivatives as in vivo imaging agents for cancers.

## 2. Materials and Methods

### 2.1. General Information

All reagents were commercial products and were used without further purification unless specified otherwise. ^67^Ga-citrate was purchased from Fujifilm RI Pharma Co., Ltd. (Tokyo, Japan). FITC was purchased from Sigma-Aldrich (St Louis, MO, USA). Fluorenylmethyloxycarbonyl (Fmoc)-ε-Ahx-OH was obtained from Merck Millipore (Billerica, MA, USA). The other Fmoc amino acids, 2-(1H-benzotriazole-1-yl)-1,1,3,3-tetramethyluronium hexafluorophosphate (HBTU) and hydroxybenzotriazole (HOBt), were purchased from Watanabe Chemical Industries Co., Ltd. (Hiroshima, Japan). Rink amide AM resin was purchased from Merck Millipore (Darmstadt, Germany). Mass spectra were obtained using MALDI-TOF-MS using Ultraflex MALDI TOF/TOF MS (Bruker Daltonics, Bremen, Germany). HPLC analysis was performed using a Shimadzu HPLC system (LC-10AT pump with SPD-10A UV detector; λ = 254 nm). A gamma survey meter (Aloka, Tokyo, Japan) was used as the refractive index (RI) detector. An automated gamma counter with an NaI (Tl) detector (2470 WIZARD2, Perkin-Elmer, Waltham, MA, USA) was used to measure radioactivity. A multi-mode reader (Cytation3; Biotek, Winooski, VT, USA) was used to measure absorbance and fluorescence.

### 2.2. Peptide Synthesis

All peptides were synthesized by a stepwise solid-phase method using *N*-9-Fmoc chemistry. Fmoc-amino acids on 100–150 mg of Rink amide AM resin with a loading capacity of 0.55 were used. The resin was first soaked in dimethylformamide (DMF) for 18 h. The Fmoc group was then deprotected using 20% (*v*/*v*) piperidine in DMF. The piperidine was then washed out, and amide coupling was carried out by adding Fmoc-amino acid (3 equivalents), HBTU/HOBt (3 equivalents), and *N*,*N*-diisopropylethylamine (DIPEA; 6 equivalents) in DMF (1.2 mL) to the reaction mixture. The Fmoc groups were then removed. For fluorescence labeling, the peptides were reacted with FITC (3 equivalents) in the presence of DIPEA (6 equivalents) in DMF at room temperature for 18 h. The introduction of the 1,4,7-triazacyclononane-1,4,7-triacetic acid (NOTA) unit at the *N*-terminal position was accomplished according to a literature-reported method [27,28]. In brief, the amino terminus of the peptide on resin was functionalized with bromoacetic acid (5 equivalents) and 1-ethyl-3-(3-dimethylaminopropyl)carbodiimide (2.5 equivalents) in dichloromethane (DCM). Then, 1,4,7-triazacyclononane (5 equivalents) in DCM was added, and the mixture was stirred for 3 h. The remaining secondary amines on the NOTA ring were substituted by reacting with tert-butyl,2-bromoacetate (3 equivalents) and DIPEA (3 equivalents) in 1-methyl-2-pyrrolidone (2 mL) for 2 h. Upon the completion of labeling, the peptide was cleaved from the resin by treatment with trifluoroacetic acid (TFA)/H_2_O/triisopropyl silane/1,2-ethanedithiol (94:2.5:1:2.5 *v*/*v*/*v*/*v*) for 90 min with shaking. After separating the peptide from the resin, the filtrate was precipitated with chilled diethyl ether. The precipitate was centrifuged at 2500 relative centrifugal force (rcf) for 5 min, washed twice with diethyl ether, and centrifuged between each washing step. The crude products were purified by HPLC on a Cosmosil C_18_ column (Nacalai Tesque, 5C_18_-AR-II, 10 × 250 mm) using a water–acetonitrile (0.1% TFA) gradient at a flow rate of 2.0 mL/min. Each peptide eluted was analyzed by MALDI-TOF-MS.

### 2.3. Synthesis of Ga-NOTA-CAPs

A solution of Ga(NO_3_)_3_·× H_2_O (5.1 mg; 20 μmol) was added to 200 µL of a solution of NOTA-CAPs in a 0.4 M 4-(2-hydroxyethyl)-1-piperazineethanesulfonic acid (HEPES) buffer (2.0 mM at pH 4.0) and heated at 95 °C for 30 min. The reaction solution was purified by HPLC, and the peptides were analyzed by MALDI-TOF-MS.

### 2.4. Synthesis of ^67^Ga-NOTA-CAPs

A solution of ^67^Ga-citrate (50 µL and 74 MBq/mL) was added to 25 µL of a solution of NOTA-CAPs in a 0.4 M HEPES buffer (2 mM ay pH 4.0) and heated at 95 °C for 30 min. The reaction mixture was purified by HPLC.

### 2.5. Preparation of Liposomes

1,2-Dimyristoyl-sn-glycero-3-phosphocholine (DMPC) and 1,2-dioleoyl-sn-glycero-3-phospho-L-serine sodium salt (DOPS) in the indicated molar ratios were dissolved in CHCl_3_ in a round-bottom flask. The solvent was gradually evaporated under reduced pressure to form a thin lipid film on the inner wall of the flask. To this, a 10 mM HEPES buffer (pH 7.4) was added, and dynamic light scattering and zeta potential were measured. Then, 50 mM bis-tris propane (BTP) and a 150 mM NaF buffer (pH 7.4) were added for circular dichroism (CD) spectroscopy. The mixture was vortexed until the thin lipid film at the bottom of the flask was completely dispersed in the buffer. The suspension was sonicated using a Sonifier 250D probe-type sonicator (Branson, Danbury, CT, USA) at 60–70 W for 25 min in an ice-water bath. The resulting liposomal suspension was centrifuged at 18,600 rcf for 1 h. The supernatant was used as the liposomal solution in subsequent steps.

### 2.6. Measurement of Zeta Potential of the CAP–Liposome Mixtures

NOTA-CAPs (10 or 20 mM) were added to each liposomal solution and incubated for 1 h at 37 °C. The dimensions and zeta (ζ) potential of the mixture were calculated using a Zetasizer Nano-ZS system (Malvern, United Kingdom) equipped with an He–Ne laser operating at a wavelength of 633 nm.

### 2.7. CD Spectroscopy of the CAP–Liposome Mixture

To each liposomal solution, 0.15 mM NOTA-CAPs in a pH 7.4 buffer (50 mM BTP; 150 mM NaF) was added. Wavelength spectra were measured from 190 to 250 nm at 37 °C using a 0.1 mm path length quartz cell. CD spectra were measured from 190 to 250 nm in a 1.0 mm path length cell using a JASCO J-725N spectropolarimeter (JASCO). Data are expressed as the residue molar ellipticity [θ]_R_ (deg·cm^2^·dmol^–1^).

### 2.8. In Vitro Stability of ^67^Ga-NOTA-CAPs in Phosphate-Buffered Saline and Mouse Plasma

The in vitro stability of ^67^Ga-NOTA-CAPs in phosphate-buffered saline (PBS) and mouse plasma was analyzed according to a previously reported procedure [29,30,31]. In brief, each radiotracer (10 μL; 2.0–3.0 MBq) was added to PBS (90 μL) and incubated at 37 °C for 24 h or to freshly prepared mouse plasma (90 μL) and incubated at 37 °C for 1 h. The PBS solution was directly injected into the HPLC system for analysis. The mouse plasma solution was added to ice-cold CH_3_CN (300 μL) and centrifuged at 9400 rcf for 4 min. The supernatant was filtered through a 0.22 μm membrane before HPLC analysis.

### 2.9. Cell Cultures

Epidermoid carcinoma KB cells were provided by DS Pharma Biomedical Co., Ltd. (Suita, Japan). Mouse fibroblast-like 3T3-L1 cells were obtained from the JCRB Cell Bank (Tokyo, Japan). KB cells were cultured in Eagle’s minimal essential medium supplemented with 10% fetal bovine serum. The 3T3-L1 cells were grown in Dulbecco’s modified Eagle’s medium supplemented with 10% bovine calf serum. All media were supplemented with 100 IU/mL penicillin and 100 μg/mL streptomycin. The cells were maintained in a humidified atmosphere containing 5% CO_2_ at 37 °C.

### 2.10. Measurement of Zeta Potential of Cell Cultures

Cultured KB cells and 3T3-L1 cells were detached from the growth surfaces using trypsin and suspended in PBS (pH 7.4) to achieve a concentration of 1.0 × 10^5^ cells/mL. The zeta potential of each cell line was measured using an ELSZ-2000ZS system (Otsuka Electronics, Tokyo, Japan).

### 2.11. Cellular Uptake Study

Each cell line was grown to confluence in a 12-well plate and washed with PBS. FITC-CAPs (1 μM) or ^67^Ga-NOTA-CAPs (1.0 kBq) (Appendix A) in a growing medium (0.6 mL) were added to each cell and incubated at 37 °C for each time point. After incubation, the cells were washed twice with 20 units of heparin/PBS (0.5 mL) and lysed using cell lysis buffer M from Fujifilm Wako (Osaka, Japan) at 37 °C for 20 min. The radioactivity and fluorescence of the lysates was measured using a gamma counter and fluorescence microplate reader, respectively, and the protein concentration was determined using a Bradford protein assay kit (Bio-Rad, Hercules, CA, USA). Data are expressed as percentage uptake per milligram of protein (% uptake/mg protein).

### 2.12. Confocal Fluorescence Microscopic Imaging of Cells Exposed to ^67^Ga-NOTA-CAPs

KB and 3T3-L1 cells were cultured on 8-well chambered cover glasses (Iwaki, Tokyo, Japan) and incubated with FITC-CAPs (10 μM) for 1 h. The medium was removed and cells were washed twice with ice-cold PBS containing heparin (20 units/mL) and twice with ice-cold PBS. Subsequently, the cells were incubated with Hoechst 33342 (2.5 µg/mL) for 30 min and then washed twice with ice-cold PBS. Fluorescent images were captured using a confocal laser scanning microscope (LSM710, Carl Zeiss Inc., Jena, Germany) with excitation wavelengths of 405 nm (UV laser) for Hoechst 33342 and 488 nm (Ar laser) for FITC-CAPs.

### 2.13. Evaluation of the Internalization Pathway of CAPs

Each cell line grown in a 12-well plate was incubated with 1.0 kBq of ^67^Ga-CAPs (0.6 mL) in the presence or absence of the internalization inhibitor at 37 or 4 °C for 1 h. The macropinocytosis inhibitor, 5-(*N*-ethyl-N-isopropyl) amiloride (EIPA) (100 µM) [32]; caveolae-dependent endocytosis inhibitor, nystatin (50 µM) [33]; or clathrin-dependent endocytosis inhibitor, sucrose (450 µM) diluted in serum-free medium was used to block each internalization. The rest of the process was performed according to the procedure described for the cell uptake study.

### 2.14. Tumor Xenograft Model

BALB/c nu/nu mice (female; 4 weeks old) were supplied by Japan SLC Inc. (Shizuoka, Japan). The mice were maintained in a room at a constant ambient temperature and a 12/12 h light/dark cycle, and they were given free access to food and water. Mice were subcutaneously injected on their right shoulders with approximately 1.0 × 10^7^ KB cells. The tumors were allowed to reach 300–500 mm^3^ (1‒2 weeks after inoculation) before biodistribution studies were performed.

### 2.15. Biodistribution of ^67^Ga-CAPs in Tumor-Bearing Mice

The ^67^Ga-NOTA-CAPs (10 kBq/100 μL) were intravenously injected into the mice via the tail vein. The mice were sacrificed at the designated time intervals, and the organs were dissected. The tissues were then weighed, and radioactivity was measured by automated gamma counting (*n* = 5).

### 2.16. Statistical Analysis

Differences in the cellular uptake of FITC-CAPs were analyzed by ANOVA with Tukey’s multiple comparison test (Figure 2). Differences in the cellular uptake of ^67^Ga-NOTA-CAPs were analyzed by ANOVA with Bonferroni’s multiple comparison test. Differences in the inhibition of the cellular uptake of ^67^Ga-NOTA-CAPs were analyzed using ANOVA with Dunnett’s multiple comparison test. Differences in tumor uptake, tumor-to-blood ratios, and tumor-to-muscle ratios of ^67^Ga-NOTA-CAPs were analyzed using ANOVA with Tukey’s multiple comparisons test in KB tumor-bearing mice. Values of *p* < 0.05 were considered statistically significant.

## 3. Results

### 3.1. In Vitro Binding of FITC-Labeled CAPs to Cancer Cells and Healthy Cells

To assess the ability of each CAP to act as a cancer diagnostic agent, we synthesized FITC-labeled CAPs via standard Fmoc solid-phase peptide synthesis. The target peptides were identified using MALDI-TOF MS (Appendix A). Cell uptake studies of FITC-labeled CAPs were performed on KB cancer cells and 3T3-L1 fibroblasts to screen potential cancer imaging agents. First, we measured the zeta potential of each cell to assess the charge on the cell surface membrane (Appendix A), and then we confirmed that the zeta potential of the KB cells (−26.6 ± 2.19 mV) was lower than that of the 3T3-L1 cells (−11.1 ± 3.13 mV). As shown in Figure 2, the uptake of FITC-KV4 in KB cells was low (5.68% ± 0.06% uptake/mg protein). FITC-KV6 exhibited the highest uptake (41.2 ± 1.97% uptake/mg protein) in KB cells in the series of peptides. The uptake of FITC-KV8 in KB cells (33.8% ± 2.47% uptake/mg protein) was slightly lower than that of FITC-KV6. FITC-KV6 showed a significantly lower uptake in 3T3-L1 cells (24.0% ± 0.69% uptake /mg protein) than in KB cells. However, there were no significant differences in the cellular uptake of FITC-KV4 and FITC-KV8 between the two cell lines. We then evaluated the cellular uptake and specificity of the KV6 analogues containing arginine or histidine in place of the lysine residues. The uptake of FITC-RV6 into KB cells was comparable to that of FITC-KV6. However, there was no significant difference between the uptake of FITC-RV6 into KB cells and 3T3-L1 cells (36.0% ± 2.63% and 26.6% ± 2.18% uptake /mg protein, respectively), indicating its low specificity for cancer cells. Though FITC-HV6 showed a lower uptake into KB cells than its two analogues (19.0% ± 0.94% uptake/mg protein), its uptake into 3T3-L1 cells was significantly lower (6.35.% ± 1.21% uptake/mg protein), thus indicating a high selectivity for cancer cells.

Confocal fluorescence imaging studies were conducted to examine the cellular localization of FITC-CAPs. FITC-KV6 predominantly accumulated on the surface and inside the cell membrane of KB cells (Figure 3A). FITC-RV6 and FITC-HV6 showed similar distributions in KB cells (Figure 3B,C, respectively), but they also accumulated in the cytoplasm and nucleus. The majority of FITC-CAPs were bound to the inner and outer surfaces of the cell membrane, consistent with the proposed mechanism of SVS-1 action [12]. The fluorescence signal of FITC-KV6 was much weaker in 3T3-L1 cells (Figure 3D). In contrast, FITC-RV6 showed strong fluorescence signals in 3T3-L1 cells, similar to those seen in KB cells (Figure 3E). FITC-HV6 showed low fluorescence signals in 3T3-L1 cells, but a large number of cells were stained (Figure 3F).

### 3.2. Zeta Potential and CD Spectra of NOTA-CAPs in the Presence of a Lipid Membrane

The macrocyclic chelator NOTA and its analogs have been reported to be the most stable chelators for radiogallium [34]. Therefore, we synthesized NOTA-CAPs for use in gallium-labelled radioligands (Figure 1B). As indicated by previous reports [16,35], NOTA-CAPs can bind, fold, and insert into lipid bilayers with surfaces rich in anionic phospholipids. Therefore, the binding interaction between NOTA-CAPs and the four types of liposomes with different phosphatidylcholine (PC)–PS ratios as biomimetic membrane models was investigated by comparing their zeta potentials. When the zeta potential of liposomes was measured, all four showed negative values. The higher the PS content of the liposome, the more negative the zeta potential was. On addition of CAPs, the zeta potential of the liposomes increased (Figure 4A–C). NOTA-KV6 and NOTA-RV6 caused dose-dependent increases in the zeta potential of the liposomes. Liposomes with a high PS content exhibited a greater increase in zeta potential (Figure 4A,B).

KV6 is thought to be internalized into cancer cells after binding to the lipid bilayer and folding into a β-sheet [12]. The structure of the peptide during interaction with the cell membrane can greatly affect its ability to accumulate in the tumor. Therefore, we evaluated the CD spectra of the CAPs in the presence or absence of model liposomes in a pH 7.4 buffer (50 mM BTP and 150 mM NaF) to assess changes in the secondary structures of the peptides. As shown in Figure 5A,B, the CD spectrum suggested that NOTA-KV6 and NOTA-RV6 exist primarily as random coils. The spectrum of these three peptides in neutral 1,2-dioleoyl-sn-glycero-3-phosphocholine (DOPC) vesicles also showed that the CAPs exist as random coils. When NOTA-KV6 or NOTA-RV6 were added to a solution containing negatively charged lipid vesicles made from a 1:1 mixture of DOPC and DOPS, the CD spectra exhibited a minimum around 218 nm, which is characteristic of the β-sheet [36]. The secondary structure of NOTA-HV6 in the buffer alone was a mixture of random coils and β-sheets. The spectrum of NOTA-HV6 did not significantly change in the presence of DOPC or DOPA/DOPS liposomes (Figure 5C). This suggested that the structural conformation of NOTA-HV6 does not change on binding to the negatively charged phospholipids at a neutral pH.

### 3.3. Radiosynthesis and In Vitro Stability of ^67^Ga-NOTA-CAPs

Next, we synthesized radiogallium-labeled NOTA-CAPs for nuclear imaging. Each NOTA-CAP was labeled with ^67^Ga (Figure 1B). The radiochemical purity of the ^67^Ga-NOTA-CAPs was tested by HPLC, and it was found to be greater than 95%. The in vitro stability was assessed by incubating the ^67^Ga-NOTA-CAPs at 37 °C in PBS and mouse plasma for 24 and 1 h, respectively (Appendix A). All the peptides showed a high stability in the PBS solution, with over 97% remaining unchanged. ^67^Ga-NOTA-KV6 was also found to be stable in mouse plasma, with 75.7% remaining unchanged. Meanwhile, only 27.4% of ^67^Ga-NOTA-RV6 and 23.3% of ^67^Ga-NOTA-HV6 remained unchanged in mouse plasma.

### 3.4. In Vitro Cellular Uptake of ^67^Ga-NOTA-CAPs

The uptake of ^67^Ga-labeled CAPs into KB and 3T3-L1 cells is shown in Figure 6. No cell growth inhibition was observed by microscopy in the presence of ^67^Ga-NOTA-CAPs (data not shown). The concentration of ^67^Ga-NOTA-KV6 (Figure 6A), ^67^Ga-NOTA-RV6 (Figure 6B), and ^67^Ga-NOTA-HV6 (Figure 6C) in KB cells reached a plateau at 60 min and maintained a high concentration until 120 min (78.8% ± 7.17%, 143% ± 11.7%, and 52.8% ± 9.37% uptake/mg protein, respectively). While the uptake of ^67^Ga-NOTA-RV6 in normal 3T3-L1 cells (131% ± 7.36% uptake/mg protein) was similar to that in KB cells, the uptake of ^67^Ga-NOTA-KV6 and ^67^Ga-NOTA-HV6 in 3T3-L1 cells (26.9% ± 2.04% and 6.51% ± 0.91% uptake/mg protein, respectively) was significantly lower than that in KB cells. These results indicate that ^67^Ga-NOTA-KV6 and ^67^Ga-NOTA-HV6 exhibit selective cellular uptake into cancer cells, while the uptake of ^67^Ga-NOTA-RV6 is non-specific. We subsequently performed additional studies of ^67^Ga-NOTA-KV6 uptake by other cancer cells (HeLa and U87-MG cells), and we observed no significant difference in the cellular uptake of the tracer among cancer cell types. The uptake of ^67^Ga-NOTA-KV6 by these three cancer cell lines was significantly higher than that of 3T3-L1 cells (Appendix A).

### 3.5. Evaluation of the Internalization Pathway for CAPs

To determine the pathway for the internalization of CAPs, we evaluated the effect of the inhibition of endocytosis on the cellular uptake of these ^67^Ga-NOTA-CAPs. KB cells were incubated with each of the peptides at 4 °C to inhibit endocytosis [37]. We also investigated the effect of the inhibition of major endocytic pathways on peptide uptake into KB cells. EIPA, nystatin, and sucrose were used to inhibit macropinocytosis- [32], caveolae- [33], and clathrin-mediated [38] endocytosis, respectively. The internalization of ^67^Ga-NOTA-KV6 was significantly suppressed by temperature reduction (72%), EIPA treatment (49%), and nystatin treatment (40%). A slight decrease in the uptake of ^67^Ga-NOTA-KV6 was also observed upon treatment with sucrose, but the difference was not statistically significant (Figure 7A). In contrast, no significant decrease in the uptake of ^67^Ga-NOTA-RV6 was observed at 4 °C or upon treatment with any of the endocytosis inhibitors (Figure 7B). The uptake of ^67^Ga-NOTA-HV6 was significantly inhibited by temperature reduction (70%), and a small decrease in uptake was observed in the presence of EIPA (*p* = 0.147). Uptake was not significantly affected by treatment with nystatin or sucrose (Figure 7C).

### 3.6. In Vivo Biodistribution Studies of ^67^Ga-NOTA-CAPs

Finally, we assessed the in vivo biodistribution of ^67^Ga-CAPs at 30, 60, and 120 min after tail vein injection in BALB/c mice bearing KB xenografts. Figure 8A shows a schematic representation of the in vivo experiments. All ^67^Ga-NOTA-CAPs exhibited a low retention in the blood and a high accumulation in the liver and/or kidneys (Figure 8). ^67^Ga-NOTA-KV6 exhibited markedly high renal accumulation and retention (118% ± 21.6% injected dose/g of tissue (% ID/g) at 120 min post-injection). Its hepatic accumulation was also high (32.6% ± 4.81% ID/g at 120 min), but accumulation in other normal organs was low (Figure 8B). ^67^Ga-NOTA-RV6 showed a lower renal accumulation (26.86% ± 3.95% ID/g at 120 min) than ^67^Ga-NOTA-KV6, but it showed higher accumulation and retention in the liver and spleen (53.8% ± 3.95% and 15.0 ±% 2.70% ID/g, respectively, at 120 min) (Figure 8C). ^67^Ga-NOTA-HV6 exhibited lower levels of accumulation in healthy tissues than other CAPs while maintaining high levels of radioactivity in the kidneys (23.0% ± 1.26% ID/g at 120 min), as shown in Figure 8D. The tumor uptake of these ^67^Ga-NOTA-CAPs was high: 4.46% ± 0.58%, 4.76% ± 1.50%, and 3.18% ± 0.36% ID/g for ^67^Ga-NOTA-KV6, ^67^Ga-NOTA-RV6, and ^67^Ga-NOTA-HV6, respectively, after 30 min (Figure 8E–G). Though the radioactivity in the tumor tissue gradually decreased, the tumor/muscle ratios reached 5.05 ± 1.51, 3.35 ± 0.64, and 4.01 ± 2.05 for ^67^Ga-NOTA-KV6, ^67^Ga-NOTA-RV6, and ^67^Ga-NOTA-HV6, respectively, after 120 min (Figure 8H–J). Next, we compared the in vivo uptake into tumors, tumor-to-blood ratios, and tumor-to-muscle ratios of ^67^Ga-NOTA-CAPs in KB tumor-bearing mice at multiple time points (Appendix A). No statistical differences in the tumor uptake of ^67^Ga-NOTA-CAPs were observed. By contrast, the tumor-to-blood ratio of ^67^Ga-NOTA-KV6 was significantly higher than those of ^67^Ga-NOTA-RV6 and ^67^Ga-NOTA-HV6 at 30 min. The tumor-to-muscle ratio of ^67^Ga-NOTA-KV6 was also significantly greater than that of ^67^Ga-NOTA-HV6 at 30 min (Appendix A).

## 4. Discussion

Several CAPs, such as SVS-1 (KV6), possess the ability to preferentially kill cancerous cells [8,39]. SVS-1 only folds at the surface of cancer cells to acquire a β-hairpin structure that allows for the preferential disruption of the cancer cell membrane [12,16,17]. In this study, we synthesized and evaluated three radiogallium-labeled SVS-1 analogues as in vivo imaging agents for the detection of malignant tissues.

We designed SVS-1 analogs with varied peptide chain lengths or lysine residues replaced by other amino acids (arginine or histidine). The preliminary studies using FITC-labeled CAPs showed that the quantitative uptake and specificity of FITC-KV6 were higher than those of peptides with different numbers of repeating KV units (KV4 and KV8), as shown in Figure 2. This indicated that changing the number of KV units in SVS-1 adversely affects the peptide’s ability to target cancer cell. Medina et al. reported that the incubation of A549 lung cancer cells in the presence of SVS-1 labeled with fluorescein via a PEG linker (5 μM) resulted in the localization of the peptide on the cell membrane after 5 min. However, the peptide gradually internalized into the cell and eventually accumulated in the nucleus after 60 min of incubation [13]. In contrast, our study showed that FITC-KV6 primarily localized on the cell surface of KB cells (Figure 3A). This discrepancy may have been due to the influence of the linker and/or differences in cell lines. The substitution of lysine with other amino acids in FITC-labeled KV6 and ^67^Ga-labeled KV6 significantly changed the cellular binding properties. The arginine derivatives expressed similar patterns of accumulation in cancer and normal cells. The histidine derivatives showed lower levels of accumulation in cancer cells than other CAPs, but they were highly selective, showing extremely low levels of accumulation in healthy cells (Figure 2, Figure 3, and Figure 6).

The zeta potential and CD spectra of the CAP-phospholipid mixtures showed that NOTA-KV6 and NOTA-RV6 possess a high affinity for negatively charged regions containing PS (Figure 5 and Figure 6). These results were consistent with previous studies reporting changes in the zeta potential of artificial lipid bilayers in the presence of SVS-1 [16]. It was observed that NOTA-HV6 had a lower binding affinity for negatively charged phospholipids in a neutral solution compared to other NOTA-CAPs. This may be attributed to the fact that the side chain of histidine, an imidazole ring, is neutral under normal physiological conditions but gets protonated in the acidic tumor microenvironment [40,41]. This protonation may promote the selective internalization of the peptide into cancer cells. FITC-NOTA-HV6 and ^67^Ga-NOTA-HV6 demonstrated significantly higher accumulation in cancer cells than in normal cells (Figure 2 and Figure 6C).

A significant difference in the cellular uptake pathways of the three ^67^Ga-NOTA-CAPs was observed (Figure 7). The results suggested that the uptake of ^67^Ga-NOTA-KV6 may be through macropinocytosis- and caveolae-mediated endocytosis (Figure 7A). Previous studies have also reported that the uptake of SVS-1-based peptides and nanoparticles into cancer cells occurs via endocytosis [13,14]. The uptake of ^67^Ga-NOTA-RV6 was unchanged by the inhibition of the three endocytosis pathways (Figure 7B). It is well known that arginine-rich cell-penetrating peptides are taken up via macropinocytosis [42]. On the other hand, CLIP6, an arginine-rich SVS-1 derivative, has been shown to be able to cross cell membranes without folding by having direct access to them [18]. Similarly, ^67^Ga-NOTA-RV6 may also be internalized by direct membrane permeation rather than endocytosis. Meanwhile, macropinocytosis and other endocytic processes may be involved in the cellular uptake of ^67^Ga-NOTA-HV6.

There have been no previous reports on the biological evaluation of radiolabeled SVS-1 derivatives, including in vivo biodistribution studies. Medina et al. studied the biodistribution of a Cy5-labeled SVS-1 derivative (Cy5-PEG-GG-SVS-1) in A549 tumor-bearing mice, and they reported tumor-to-skin fluorescence ratios of approximately 2.3 within 2 h of injection [13]. In the context of their results, our study showed that of the tested peptides, ^67^Ga-NOTA-CAPs attained the highest tumor/blood ratio (3.8–8.0) and tumor/muscle ratio (3.3–5.0). It is interesting to note that while the data from in vitro studies suggested that ^67^Ga-NOTA-RV6 may not be selective towards cancer cells, in vivo distribution studies showed that the peptide accumulated to a greater extent in cancer tissue than in healthy muscle tissue. This discrepancy may be due to the in vivo tumor microenvironment; it has been reported that cell membranes bearing surface anionic phospholipids are abundant not only in tumor cells but also in the blood vessels connected to cancer tissues in tumor-bearing mice [43]. ^67^Ga-NOTA-HV6 showed lower levels of tumor uptake than the other peptides, but it had low levels of uptake into peripheral organs. It exhibited a much lower renal uptake (2.29% ± 3.95% ID/g) than ^67^Ga-NOTA-KV6 (32.6% ± 4.81% ID/g) and ^67^Ga-NOTA-RV6 (53.8% ± 3.95% ID/g) at 120 min. Sarko et al. studied the biodistribution of several radiolabeled typical cell-penetrating peptides (CPPs), including penetratin, TAT, and R_9_ peptides, in tumor-bearing mice. All of these CPPs had a poor tumor uptake, with most peptides exhibiting tumor/blood or tumor/muscle ratios of less than 1 [44]. While both ^67^Ga-NOTA-CAPs and the above CPPs are rich in basic and lipophilic amino acids, the ^67^Ga-NOTA-CAPs have a much higher tumor tissue selectivity than typical CPPs. On the other hand, the tumor/muscle ratios of tumor receptor-targeting peptides, including the α_v_β_3_ integrin-derivative ^68^Ga-c(RGDyK) [45] and the CXCR4-derivative ^68^Ga-CPCR4-2 [46] are higher (9.30 ± 3.90 and 16.6 ± 3.80, respectively, at 60 min) than that of ^67^Ga-NOTA-KV6 (5.05 ± 1.51 at 120 min). These cancer-associated protein-targeting radiotracers exhibit strong receptor-ligand binding and/or receptor-mediated internalization. On the other hand, the low retention of ^67^Ga-NOTA-CAPs in tumor tissues may be due to the enzymatic degradation and/or blood flow-mediated dissociation of radiotracers from the cancer cell membrane. In fact, the levels of authentic ^67^Ga-NOTA-CAPs decreased in the presence of mouse plasma (Appendix A), although these radiotracers are more stable in plasma than other lysine- and arginine-rich peptides, such as TAT peptide (half-life = 3.3 min) [47]. The metabolic stability of these radiotracers could be improved by substituting the l-amino acid residues with d-amino acids or other unnatural amino acids. Glycoprotein carbohydrates rich in phosphatidylserine and sialic acid represent one possible binding target for ^67^Ga-NOTA-CAPs. Their targeted moiety may be completely different from that of all imaging probes currently under clinical development that target proteins that are highly expressed in cancer tissues. These CAPs may thus serve as vehicles that can enhance the functionality of previously developed tumor imaging agents. In future studies, hybrid molecules comprised of radiolabeled NOTA-CAPs with cancer-targeting peptides, anticancer agents, therapeutic radionuclides, and/or nanoparticles are likely to be used for cancer theranostics, including nuclear imaging and internal radiation therapy.

## 5. Conclusions

In this study, we developed three ^67^Ga-NOTA-CAPs, of which two exhibited selective accumulation in cancer cells. These peptides show significantly different mechanisms for interaction with artificial lipid bilayers and cell internalization. In vivo distribution studies found that all three ^67^Ga-NOTA-CAPs showed good tumor uptake, tumor-to-blood ratios, and tumor-to-muscle ratios. These CAPs have the ability to target cancer tissues in vivo, and they are promising scaffolds for potential theranostic agents.

## Figures and Tables

**Figure 1 cancers-13-02388-f001:**
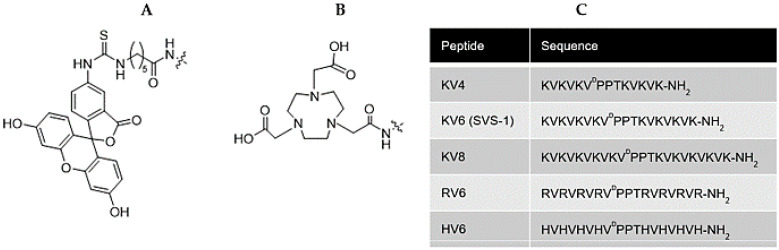
Design of fluorophore- and radiogallium chelator-conjugated cationic amphiphilic peptides (CAPs) as tumor imaging agents. Fluorescein- (**A**) and NOTA- (**B**) conjugated CAPs consisting of two segments of repeated cationic amino acid (lysine, arginine, or histidine) and valine residues joined by the β-turn tetrapeptide (−V^D^PPT−, where ^D^P is d-proline), which allows the peptides to form folded β-hairpin structures on interaction with cancer cell membranes. Five peptide sequences were used as CAPs in this study (**C**).

**Figure 2 cancers-13-02388-f002:**
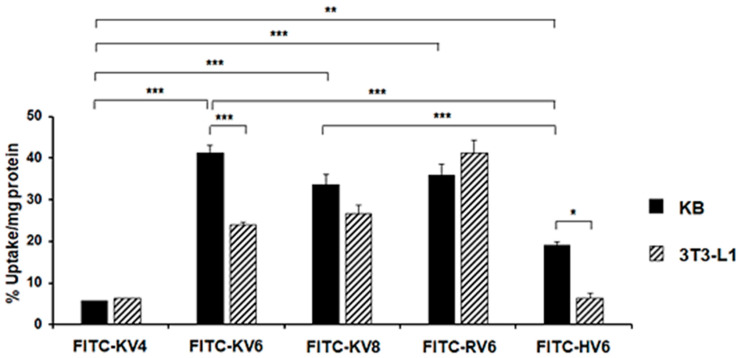
Cellular uptake of FITC-CAPs into KB and 3T3-L1 cells at 2 h post-incubation. * *p* < 0.005, ** *p* < 0.0005, and *** *p* < 0.0001 (ANOVA and Tukey’s *t* test). Values are means ± SEM; *n* = 7–13.

**Figure 3 cancers-13-02388-f003:**
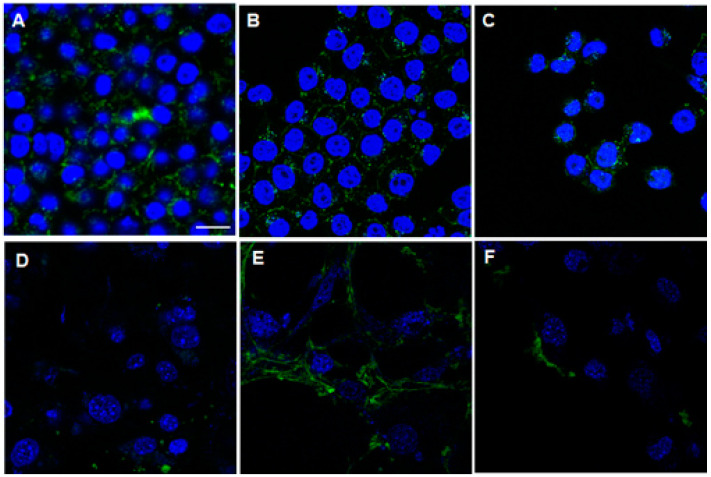
Conforcal fluorescence imaging of FITC-CAPs in cell lines. KB (**A**–**C**) and 3T3-L1 (**D**–**F**) cells treated with 10 µM FITC-CAPs (green) for 1 h. Nuclei were stained by Hoechst 33342 (blue). Fluorescence images were taken with a confocal laser scanning microscope. Scale bar = 20 μm.

**Figure 4 cancers-13-02388-f004:**
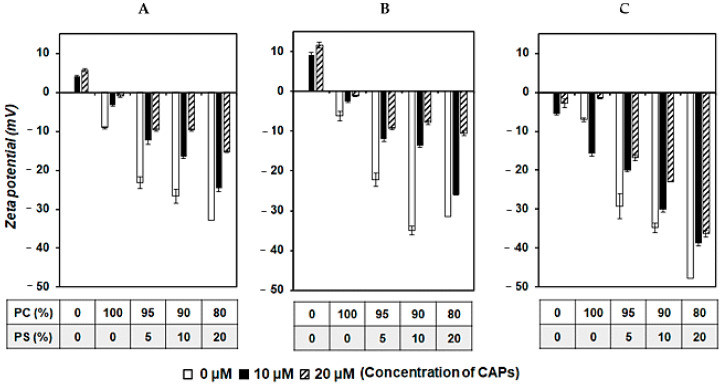
Zeta potential of liposomes composed of different PC/PS ratios in the presence or absence of NOTA-CAPs in a 10 mM HEPES buffer (pH 7.4). Liposomes were treated with 0, 10, or 20 µM of NOTA-KV6 (**A**), NOTA-RV6 (**B**), or NOTA-HV6 (**C**) for 1 h. Values are means ± SEM; *n* = 3.

**Figure 5 cancers-13-02388-f005:**
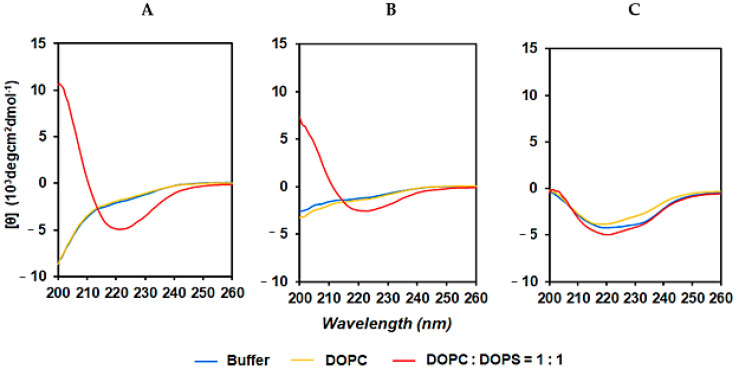
CD spectra of NOTA-CAPs in the presence and absence of liposomes. CD spectra of 150 μM NOTA-KV6 (**A**), NOTA-RV6 (**B**), and NOTA-HV6 (**C**) in an aqueous buffer solution (50 mM BTP and 150 mM NaF at pH 7.4) (blue line) in the presence of neutral PC liposomes (yellow line) and in the presence of negatively charged DOPC/DOPS liposomes (1:1) (red line).

**Figure 6 cancers-13-02388-f006:**
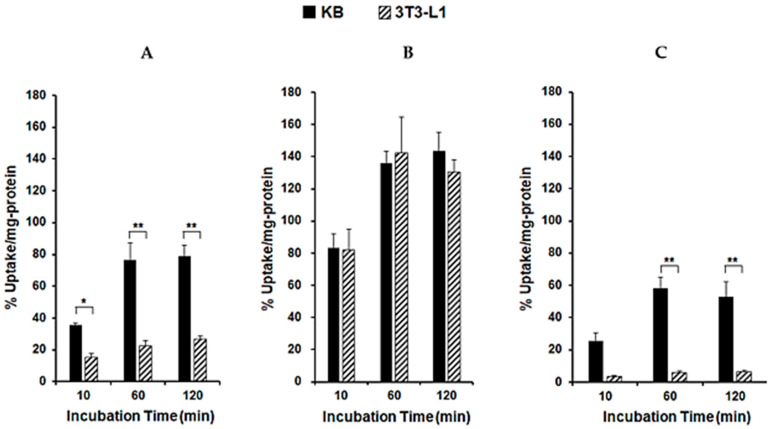
Cellular uptake of ^67^Ga-NOTA-KV6 (**A**), ^67^Ga-NOTA-RV6 (**B**), and ^67^Ga-NOTA-HV6 (**C**) in KB and 3T3-L1 cells. * *p* < 0.01 and ** *p* < 0.001 compared with 3T3-L1 cells. (ANOVA Bonferroni’s multiple comparison test). Values are mean ± SEM (*n* = 4–6).

**Figure 7 cancers-13-02388-f007:**
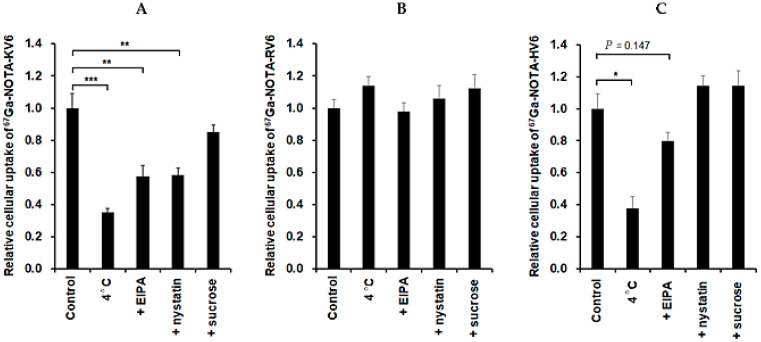
Cellular uptake inhibition of ^67^Ga-NOTA-KV6 (**A**), ^67^Ga-NOTA-RV6 (**B**), and ^67^Ga-NOTA-HV6 (**C**) for 60 min in KB cells. Cells were incubated at 4 °C to investigate energy-dependent uptake. Three different inhibitors: a macropinocytosis inhibitor (EIPA), a caveolae inhibitor (nystatin), and a clathrin inhibitor (sucrose) were used to investigate the effects of inhibition of endocytosis. * *p* < 0.001 ** *p* < 0.005, and *** *p* < 0.0001 compared with control (ANOVA Dunnett’s multiple comparison test). Values are mean ± SEM (*n* = 12).

**Figure 8 cancers-13-02388-f008:**
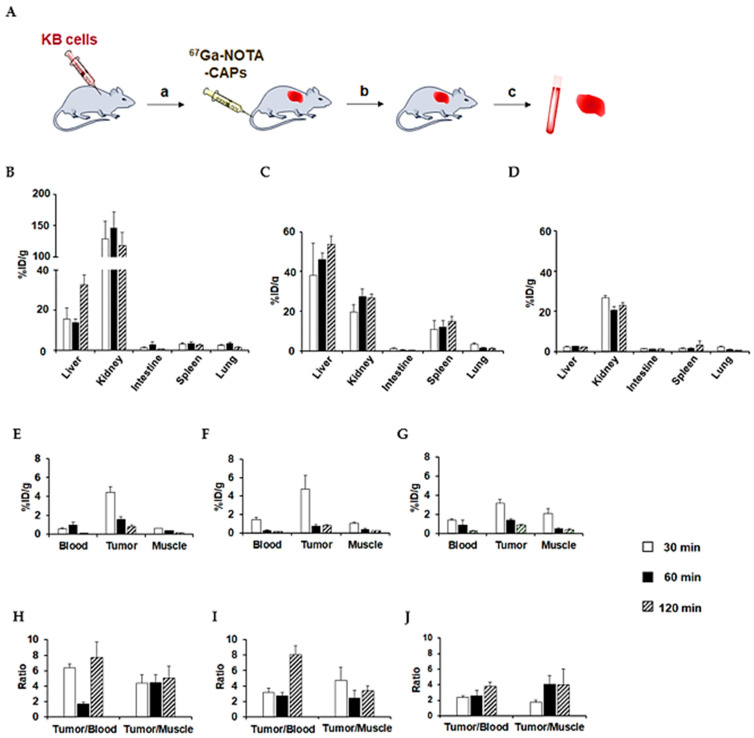
Schematic of the experimental design of the in vivo experiments (**A**): BALB/c nu/nu mice were subcutaneously injected with 1.0×10^7^ KB cells (**a**). Tumors were allowed to reach 300–500 mm^3^ (1‒2 weeks after inoculation) prior to conducting bio-distribution studies (**b**). ^67^Ga-NOTA-CAPs (10 kBq/100 μL) were intravenously injected via the tail vein (**c**). Mice were sacrificed at each time point, and then blood samples were taken and organs were dissected. Tissues were then weighed and radioactivity was measured by automated gamma counting. In vivo biodistribution of ^67^Ga-NOTA-KV6 (**B**,**E**), ^67^Ga-NOTA-RV6 (**C**,**F**), and ^67^Ga-NOTA-HV6 (**D**,**G**) over time in KB tumor-bearing mice. Data are expressed as percent injected dose per gram of tissue (%ID/g). Each value represents the mean (standard deviation) of 4–6 mice. Tumor-to-blood and tumor-to-muscle ratios for ^67^Ga-NOTA-KV6 (**H**), ^67^Ga-NOTA-RV6 (**I**), and ^67^Ga-NOTA-HV6 (**J**) were calculated at different time points.

## Data Availability

Data is contained within the article or Appendix A.

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
