# Peer review of "Synthesis and Characterization of Radiogallium-Labeled Cationic Amphiphilic Peptides as Tumor Imaging Agents"

_cancers, 2021, doi:10.3390/cancers13102388_

Round 1
Reviewer 1 Report
In this report, the authors describe the synthesis and characterization of SVS-1 cationic amphiphilic peptides (that exhibit antitumor activity) composed of 6 KV repeating lysine-valine residues, as well as with 4 and 8 KV repeats. They also have designed novel peptides in which the lysine residues of KV6 have been replaced by arginine or histidine (RV6 and HV6), and they have used 67Ga for radiolabeling. In this work, they have synthesized and evaluated some fluorescein isothiocyanate isomer I (FITC)- and 67Ga-labeled KV6 derivatives as in vivo imaging agents for cancers (67Ga-NOTA-KV6, 67Ga-NOTA-RV6 and 67Ga-NOTA-HV6). Two of them exhibited selective accumulation in cancer cells, and in vivo distribution studies showed that the three 67Ga-NOTA-CAPs showed good tumor uptake, tumor-to-blood ratios and tumor-to-muscle ratios. Therefore, the CAPs of this work are promising scaffolds for potential theranostic agents.
The paper is interesting and a lot of work has been developed. Although I was expecting some PET and/or SPECT images, I recommend its publication in Cancers.
I only have a couple of points to be improved and three mistakes:
- Can the authors add a synthesis scheme to be more clarifying?
- Can the authors expand a little bit the confocal fluorescence imaging studies section, to better explain the localization of FITC-CAPs?
- In Figure 1 C there is not any structure drawn.
- Page 4 line 136, “thrice” I guess that is twice.
- Page 6 line 252, “3V3-L1 cells” I assume these are cells 3T3-L1.
Author Response
Response to Reviewer 1 Comments
Overall comment: In this report, the authors describe the synthesis and characterization of SVS-1 cationic amphiphilic peptides (that exhibit antitumor activity) composed of 6 KV repeating lysine-valine residues, as well as with 4 and 8 KV repeats. They also have designed novel peptides in which the lysine residues of KV6 have been replaced by arginine or histidine (RV6 and HV6), and they have used 67Ga for radiolabeling. In this work, they have synthesized and evaluated some fluorescein isothiocyanate isomer I (FITC)- and 67Ga-labeled KV6 derivatives as in vivo imaging agents for cancers (67Ga-NOTA-KV6, 67Ga-NOTA-RV6 and 67Ga-NOTA-HV6). Two of them exhibited selective accumulation in cancer cells, and in vivo distribution studies showed that the three 67Ga-NOTA-CAPs showed good tumor uptake, tumor-to-blood ratios and tumor-to-muscle ratios. Therefore, the CAPs of this work are promising scaffolds for potential theranostic agents.
The paper is interesting and a lot of work has been developed. Although I was expecting some PET and/or SPECT images, I recommend its publication in Cancers.
Response: We appreciate the positive comments from the reviewer.
Point 1: Can the authors add a synthesis scheme to be more clarifying?
Response 1: We thank the reviewer for the suggestion. As suggested by the reviewer, we added detailed synthetic scheme of FITC-CAPs and NOTA-CAPs as scheme S1 in the supplementary material file.
Point 2: Can the authors expand a little bit the confocal fluorescence imaging studies section, to better explain the localization of FITC-CAPs?
Response 2: We thank the referee for the comment. As suggested by the reviewer, we have added the above responses in the result section of the revised manuscript, as follows:
<<The majority of FITC-CAPs were bound to the inner and outer surfaces of the cell membrane, consistent with the proposed mechanism of SVS-1 action [12].>>

Reviewer 2 Report
The manuscript titled “Synthesis and Characterization of Radiogallium-labeled Cationic Amphiphilic Peptides as Tumor Imaging Agents” by Fuchigami et al. describes the discovery of a theranostic radiotracer agent. The authors developed radio-gallium labeled Ga-NOTA-KV6, Ga-NOTA-RV6, Ga-NOTA-HV6 radiotracers for cancer theranostics. Published data suggest that SVS-1 only drives its own β-folding on the tumor cell surface, where it then forms pores to destroy the cancer cell. Gallium Ga 68-labeled DOTA Di-HSG peptide IMP-288 are also published. The authors have done a lot of experiments to show the importance of Ga-NOTA-derivatives in this paper.
Major comments
- Please include labels to show which bar is for KB and 3T3-L1, respectively, in Figure 2. Without this information, the data cannot be interpreted.
- It will be interesting to know if the constructed radiotracers show any cytotoxic effect in vitro?
- The authors should put statistical analysis for Figure 8 as the values are too close.
- The authors may show some in vivo images in Figure 8, showing the targeting efficacy. What controls are used?
Minor comments:
- Figure 3. The authors can include details on top of each figure for quick understanding.
- Figure 5. The authors may discuss how liposomes prepared by positive and neutral lipids may behave.
- Authors may provide a schematic representation of the in vivo experimental plan for better understanding.
- Some current references may be included.
Author Response
Response to Reviewer 2 Comments
Overall comment: The manuscript titled “Synthesis and Characterization of Radiogallium-labeled Cationic Amphiphilic Peptides as Tumor Imaging Agents” by Fuchigami et al. describes the discovery of a theranostic radiotracer agent. The authors developed radio-gallium labeled Ga-NOTA-KV6, Ga-NOTA-RV6, Ga-NOTA-HV6 radiotracers for cancer theranostics. Published data suggest that SVS-1 only drives its own β-folding on the tumor cell surface, where it then forms pores to destroy the cancer cell. Gallium Ga 68-labeled DOTA Di-HSG peptide IMP-288 are also published. The authors have done a lot of experiments to show the importance of Ga-NOTA-derivatives in this paper.
Response: Thank you very much for your recognition of this paper.
Major comments
Point 1: Please include labels to show which bar is for KB and 3T3-L1, respectively, in Figure 2. Without this information, the data cannot be interpreted.
Response 1: We thank the reviewer for the suggestion. As suggested by the reviewer, we added the labels to show which bar is for KB and 3T3-L1 cells in the revised Figure 2.
Point 2: It will be interesting to know if the constructed radiotracers show any cytotoxic effect in vitro?
Response 2: We appreciate the reviewer for this suggestion. Although we have not performed cell viability and cell proliferation, there are no cell growth inhibition in the presence of 67Ga-NOTA-CAPs observed by microscope.
We have added the above responses in the result section of the revised manuscript, as follows:
<<No cell growth inhibition was observed by microscopyin the presence of 67Ga-NOTA-CAPs (data not shown).>>
Point 3: The authors should put statistical analysis for Figure 8 as the values are too close.
Response 3: We thank the referee for the comment. As suggested by the reviewer, we added Figure S2 which compares in vivo tumor uptake, tumor-to-blood ratios, and tumor-to-muscle ratios of 67Ga-NOTA-CAPs in KB tumor-bearing mice at each time point. The results show that there are no statistical differences among tumor uptake of 67Ga-NOTA-CAPs. On the other hand, tumor-to-blood ratio of 67Ga-NOTA-KV6 was significantly higher than those of 67Ga-NOTA-RV6 and 67Ga-NOTA-HV6 at 30 min. The tumor-to-muscle ratio of 67Ga-NOTA-KV6 was also significantly greater compared with 67Ga-NOTA-HV6 at 30 min.
We have incorporated the above responses in the result section of the revised manuscript, as follows:
<<Next, we compared in vivo tumor uptake, tumor-to-blood ratios, and tumor-to-muscle ratios of 67Ga-NOTA-CAPs in KB tumor-bearing mice at each time point. The results show that there are no statistical differences among tumor uptake of 67Ga-NOTA-CAPs. On the other hand, tumor-to-blood ratio of 67Ga-NOTA-KV6 was significantly higher than those of 67Ga-NOTA-RV6 and 67Ga-NOTA-HV6 at 30 min. The tumor-to-muscle ratio of 67Ga-NOTA-KV6 was also significantly greater compared with 67Ga-NOTA-HV6 at 30 min (Figure S2).>>
Minor comments:
Point 1: Figure 3. The authors can include details on top of each figure for quick understanding.
Response 1: We appreciate the reviewer for this suggestion. As suggested by the reviewer, we did not fully explain the experimental operation on this data. Therefore, we have incorporated the above responses in the materials and methods section and Figure 3 of the revised manuscript, as follows:
Materials and methods:
<< 2.12. Confocal fluorescence microscopic imaging of cells exposed to 67Ga-NOTA-CAPs
KB and 3T3-L1 cells were cultured on 8-well chambered cover glasses (Iwaki, Tokyo, Japan) and incubated with FITC-CAPs (10 μM) for 1 h. The medium was removed and cells were washed twice with ice-cold PBS containing heparin (20 units/mL) and twice with ice-cold PBS. Subsequently, the cells were incubated with Hoechst 33342 (2.5 µg/mL) for 30 min, then washed twice with ice-cold PBS. Fluorescent images were captured using a confocal laser scanning microscope (LSM710, Carl Zeiss Inc., Jena, Germany) with excitation wavelengths of 405 nm (UV laser) for Hoechst 33342, and 488 nm (Ar laser) for FITC-CAPs. >>
Figure 3:
<< Figure 3. KB (A−C) and 3T3-L1 (D−F) cells treated with 10 µM FITC-CAPs (green) for 1 h. Nuclei were stained by Hoechst 33342 (blue). Fluorescence images were taken with a confocal laser scanning microscope. Scale bar = 20 μm. >>
Point 2: Figure 5. The authors may discuss how liposomes prepared by positive and neutral lipids may behave.
Response 2: As indicated by the previous reports, NOTA-CAPs could be bound, folded and inserted at anionic phospholipid-rich lipid bilayer surface. In accordance with the reviewer’s suggestion, we added the following sentence in the result of revised manuscript.
<< As indicated by previous reports [16, 35], NOTA-CAPs can bind, fold, and insert into lipid bilayers with surfaces rich in anionic phospholipids.>>
Point 3: Authors may provide a schematic representation of the in vivo experimental plan for better understanding.
Response 3: As suggested by the reviewer, we have incorporated the schematic experimental design of in vivo experiments in the Figure 8A of revised manuscript, as follows:
<< Figure 8. Schematic of the experimental design of the in vivo experiments (A): BALB/c nu/nu mice were subcutaneously injected with 1.0×107 KB cells (a). Tumors were allowed to reach 300–500 mm3 (1‒2 weeks after inoculation) prior to conducting bio-distribution studies (b). 67Ga-NOTA-CAPs (10 kBq/100 μL) were injected intravenously via the tail vein (c). Mice were sacrificed at each time point, blood samples were taken, and organs were dissected. Tissues were then weighed and radioac-tivity was measured by automated gamma counting (d).~~ >>
Point 4: Some current references may be included.
Response 4: As suggested by the reviewer, we cited a recent paper on the histamine-containing peptide 68Ga-DOTADi-HSG peptide IMP-288.
We have added the above responses in the introduction section of the revised manuscript, as follows:
<< The histamine-containing peptide 68Ga-IMP-288 has been used for pretargeted PET imag-ing of carcinoembryonic antigen, which exhibited quite low uptake in normal tissues [20, 21]. >>

Reviewer 3 Report
This manuscript reports the study of 67Ga-labeled cationic amphiphilic peptides as tumor imaging agents. The study includes the chemical synthesis, radiolabeling, in vitro cell studies, and in vivo studies in small animal models. Though the study provides some interesting information, there are some major concerns.
- Based on the data presented, there is not enough evidence to strongly support imaging tumor through targeting cell surface negative charge is a valid approach.
- It would be important to evaluate the most promising probe in multiple tumor cell lines and animal models to validate the targeting mechanisms, specificity, and broad use of the probe.
- The in vivo performance of the probes develop is not so great.
Author Response
Response to Reviewer 3 Comments
Overall comment: This manuscript reports the study of 67Ga-labeled cationic amphiphilic peptides as tumor imaging agents. The study includes the chemical synthesis, radiolabeling, in vitro cell studies, and in vivo studies in small animal models. Though the study provides some interesting information, there are some major concerns.
Response: Thank you very much for your recognition of this paper.
Major comments
Point 1: Based on the data presented, there is not enough evidence to strongly support imaging tumor through targeting cell surface negative charge is a valid approach.
Response 1: We thank the reviewer for the suggestion. The tumor accumulation of these 67Ga-NOTA-CAP showed high levels of 3.18-4.46% ID/g at 30 min after injection (Figure 8). Although their tumor uptake decreased gradually, these tracers attained the tumor/blood ratio tumor/blood (3.8−8.0) and tumor/muscle ratios (3.3−5.0) at 2 h after administration. A possible target site for 67Ga-NOTA-CAP is a glycoprotein sugar chain rich in phosphatidylserine and sialic acid. Their target site may be completely different from the imaging probe for proteins that are highly expressed in cancer tissues currently under clinical development. Therefore, these CAPs could serve as vehicles that can enhance the functionality of previously developed tumor imaging agents. As mentioned in the discussion section, hybrid molecules comprised of radiolabeled NOTA-CAPs with cancer-targeting peptides, anticancer agents, therapeutic radionuclides, and/or nanoparticles, are likely to be used for cancer theranostics, including nuclear imaging and internal radiation therapy in the future.
We have added the above responses in the discussion section of the revised manuscript, as follows:
<<Glycoprotein carbohydrates rich in phosphatidylserine and sialic acid represent one pos-sible binding target for 67Ga-NOTA-CAPs. Their targeted moiety may be completely different from that of all imaging probes currently under clinical development that target proteins highly expressed in cancer tissues. These CAPs may thus serve as vehicles that can enhance the functionality of previously developed tumor imaging agents.>>
Point 2: It would be important to evaluate the most promising probe in multiple tumor cell lines and animal models to validate the targeting mechanisms, specificity, and broad use of the probe.
Response 2: We thank the referee for the comment. As suggested by the reviewer, we performed additional cell uptake studies of 67Ga-NOTA-KV6 in HeLa cells and U87-MG cells (Figure S2 of revised manuscript). The results showed that there is no significant difference in the cellular uptake of the tracer among the cancer cells. In addition, uptake of 67Ga-NOTA-KV6 by these three cancer cell lines was significantly higher than that of 3T3-L1 cells.
We have added the above responses in the Figure S2 and result section of the revised manuscript, as follows:
<< We subsequently performed additional studies of 67Ga-NOTA-KV6 uptake by other cancer cells (HeLa and U87-MG cells), and observed no significant difference in cellular uptake of the tracer among cancer cell types. Uptake of 67Ga-NOTA-KV6 by these three cancer cell lines was significantly higher than that of 3T3-L1 cells (Figure S2).>>
Point 3: The in vivo performance of the probes develop is not so great.
Response 3: We appreciate the reviewer for this comment. As suggested by the reviewer, the in vivo performance of these CAPs may be not so great because low retention in the blood and relatively rapid washout from the tumor tissues. As mentioned in the discussion, tumor retention could be improved using hybrid molecules comprised of radiolabeled NOTA-CAPs with cancer-targeting peptides. Blood retention may be improved by the development of a highly metabolically stable radiotracer.

Round 2
Reviewer 3 Report
The revised manuscript is acceptable for publication.
Author Response
Overall comment: The revised manuscript is acceptable for publication.
Response: We appreciate the positive comments from the reviewer.